# OpenReview forum: "TRACE: Coarse-to-Fine Automated Evaluation of Mobile Agents with Safety Considerations in Realistic Environments"
_ICLR.cc/2026/Conference — ICLR 2026 Conference Withdrawn Submission_

### Official Review · Reviewer_m47q · 2025-10-27

**Soundness:** 2
**Presentation:** 2
**Contribution:** 3
**Rating:** 2
**Confidence:** 4

**Summary:**

The paper proposes TRACE, a coarse-to-fine automated evaluation framework for mobile GUI agents that integrates step-level reasoning with task-level verification. It introduces TRACEBench, a benchmark of 187 real-world tasks across 35 apps with safety-aware and multi-modal evaluation metrics. Experiments show that TRACE achieves evaluations closely matching human judgments and outperforms prior automatic evaluators such as SPA-Bench.

**Strengths:**

1. The paper tackles an important and underexplored problem: evaluating GUI-based agents, which involves unique challenges beyond standard LLM-as-judge settings.
2. The proposed coarse-to-fine evaluation framework is novel, improving upon prior rule-based or single-VLM evaluators through structured step-level and task-level reasoning.

**Weaknesses:**

1. The overall focus of the paper is somewhat unclear. The main contribution appears to be developing a new evaluation framework for assessing the trajectories of GUI agents, which could help benchmark existing agents or assist in training stronger ones. However, the description of the proposed method is too brief and lacks sufficient clarity. It is difficult to understand how the method actually works and why it performs better than previous approaches. This lack of detail also affects the design of experiments and the explanatory figures.

2. The method section is overly simple and provides few technical details. Key implementation components are missing, making it hard to grasp how the framework operates in practice. Moreover, the illustration figures fail to convey useful information. Although the paper emphasizes a “coarse-to-fine” evaluation strategy, it does not clearly explain how step-wise information is incorporated into the final evaluation process.

3. While the paper presents a large number of experiments, the experimental focus seems misplaced. If the main goal is to validate the effectiveness of the evaluator, a large portion of the experiments (e.g., Table 1) is spent on evaluating different GUI agents, which is not the main point of the paper. At least this experiment should not be presented as the most important result.

4. The paper missed discussion with previous works on evaluting GUI agents in the related work section.

**Questions:**

1. If all step-level evaluations are run in parallel, how does each step obtain information from the previous ones? For instance, an agent may collect intermediate results or context in earlier steps, and later actions could depend on those results in ways that are not explicitly related to the task description. Similarly, an agent might perform recovery behaviors to fix earlier mistakes, which are also not directly tied to the original goal. How does TRACE handle such dependencies when evaluating each step independently?

2. How does TRACE automatically generate task milestones? In cases where a task can be completed in multiple valid ways (e.g., checking the weather through different apps or directly searching online), how does TRACE ensure that alternative valid solutions are not penalized?

3. TRACEBench contains a relatively small number of tasks compared to prior benchmarks. Given that this paper focuses on evaluation rather than agent training, why did the authors not use existing benchmarks such as AiTW to demonstrate TRACE’s effectiveness and generality?

4. Why are timeout and safety issues explicitly analyzed in the experimental section? These aspects seem to relate more to agent design or behavior, rather than to the evaluation framework itself. Clarifying the motivation behind including these metrics would help.

5. The experimental description for Table 1 is somewhat unclear. Are recall and precision defined by comparing the agent’s self-reported evaluation results with TRACE’s evaluation, or between TRACE’s evaluation and the human-annotated ground truth? If it is the former, how does this result demonstrate the effectiveness of TRACE as an evaluator?

---

> ### Author Response · Authors · 2025-11-25
>
> We thank the reviewer for the constructive comment. Next, we will clarify the comments point by point.
>
> **Weakness 1**
>
> > The overall focus of the paper is somewhat unclear. The description of the proposed method is too brief and lacks sufficient clarity.
>
>
> We apologize that the current version of the paper may have given the impression that the main focus and contributions are somewhat unclear. We would like to first clarify the three primary contributions of our work and then explain the proposed method.
>
>
> **First**, the core contribution of our work is the proposal of a new VLM-based evaluator, TRACE. Such an accurate and automated evaluator enables large-scale assessment of mobile agents and identify their respective limitations. The **second** contribution is the proposed TRACEBench consisting of 187 common tasks and the metric system. TRACEBench focuses on high-frequency tasks in everyday apps. Its distinguishing features are that it closely reflects real-world usage scenarios and considers safety issues like unauthorized orders or payments when designing tasks. The **third** contribution lies in key insights gained from evaluating eight representative mobile agents with TRACE on TRACEBench.
>
>
> The core idea of TRACE is to evaluate agent's execution trajectories  at both the step-wise and overall levels. In the step-wise assessment, the input consists of each individual action taken by the agent along with the corresponding screenshot, and the VLM evaluates the effect of each action and the associated screen clues independently. During the overall judgement stage, the VLM first generates milestones from the task description. It then integrates **the step-wise results**, **the generated milestones**, and **the final screenshot** to determine whether the agent has completed all milestones according to the requirements. This evaluation approach partially simulates how humans assess task completion and improves the accuracy of the assessment.
>
> Regarding methodological details, we have provided a detailed description of TRACE’s workflow in `Section 3`. `Figure 2` illustrates both the Step-wise Assessment and Overall Judgement processes, and `Appendix B` includes the prompts used by TRACE. We believe these details are sufficient for readers to understand how TRACE operates in practice, as TRACE is not a highly complex framework. We also acknowledge that the paper lacks a more detailed ablation study for TRACE, which may have made it difficult to understand why it outperforms previous approaches, and we will present the corresponding ablation results and detailed analyses below.

---

> > ### Author Response · Authors · 2025-11-25
> >
> > *TRACE ablation.* For the ablation study, we ablate TRACE into four variants: (i) *TRACE without milestone* (no milestone information is provided in TRACE overall judgement), (ii) *Final screen* (only using final screen for judgement), (iii) *Trajectory* (all screenshots, actions, and milestone are provided in a single query), and (iv) *Trajectory without milestone* (all screenshots and actions are provided in a single query). Each variant adopts minimal modifications on prompt template to maintain maximal similarity to TRACE. We evaluate these variants with both Qwen2.5-VL-72B-Instruct and doubao-1.5-vision-pro-250328, and the results are shown in below table. Compared with *Final screen* and *Trajectory*, TRACE shows superior evaluation performance with both two VLMs, which demonstrates the effectiveness of TRACE's design decisions. *Final screen* schema doesn't provide enough information for trajectory evaluation. *Trajectory* schema inevitably induces very long and complex contexts, which makes it difficult for VLMs to accuratly understand and evaluate the trajectory. This schema is also not suitable for very long trajectory due to the context window limitaion. As for the milestone information used in TRACE, its impact depends on the chosen VLM evaluator. For Qwen, incorporating milestone information leads to a slightly positive performance gain, whereas for Doubao the effect is negative. However, the overall performance fluctuation remains small for both models. We recommend that when a stronger, closed-source VLM is used as the evaluator, providing milestones is unnecessary; however, when using a relatively weaker VLM, such as the open-source Qwen, incorporating milestones can improve evaluation accuracy.
> > | Evaluator | VLM | Accuracy | Precision | Recall | F1-score |
> > |:---------|:---------|---------:|---------:|---------:|---------:|
> > | TRACE  | Qwen2.5-VL-72B-Instruct  | **0.855**  |  **0.741**  |  **0.958**  | **0.836**  |
> > | TRACE w/o milestone  | Qwen2.5-VL-72B-Instruct  | 0.823  |  0.710  |  0.917  |  0.800  |
> > | Final screen | Qwen2.5-VL-72B-Instruct  |  0.774  |  0.667  |  0.833  |  0.741  |
> > | Trajectory | Qwen2.5-VL-72B-Instruct  | 0.613  |  0.500  |  **0.958** |  0.657  |
> > | Trajectory w/o milestone | Qwen2.5-VL-72B-Instruct  |  0.565  |  0.469  |  **0.958**  |  0.630  |
> > | TRACE  | doubao-1.5-vision-pro-250328  | 0.903  |  0.909  |  0.833  | 0.870  |
> > | TRACE w/o milestone  | doubao-1.5-vision-pro-250328  | **0.935**  |  **0.955**  |  **0.875**  |  **0.913**  |
> > | Final screen | doubao-1.5-vision-pro-250328  |  0.710  |  0.688  |  0.458  |  0.550  |
> > | Trajectory | doubao-1.5-vision-pro-250328  | 0.790  |  0.720  |  0.750  |  0.734  |
> > | Trajectory w/o milestone | doubao-1.5-vision-pro-250328  |  0.806  |  0.773  |  0.708  |  0.739  |
> >
> > *Table R1: Ablation study of TRACE.*

---

> ### Author Response · Authors · 2025-11-25
>
> **Weakness 2**
>
> > The method section is overly simple and provides few technical details.
>
> Thanks for this valuable feedback. We acknowledge that the Method section is relatively simple, but TRACE is indeed a simple yet effective evaluator. We have provided method details in the submitted manuscript. In `Section 3.2`, we have described how step-level assessments are incorporated into the final evaluation process, which can be specifically seen in `lines 214–215` of the manuscript.
>
> In Figure 2, we provide an example showing the agent’s executed action log and corresponding screenshots. It illustrates the results of the step-wise assessment for each step, which constitute part of the input for the overall judgement, and also demonstrates task milestone decomposition and the results of the judgement in the overall judgement stage. The methodological workflow is already presented in Figure 1 and `Section 3`, while Figure 2 mainly shows an example to display the intermediate results produced by TRACE during the evaluation process.
>
> We will revise Figure 2 in the revised manuscript to more clearly demonstrate how the results of the step-wise assessment are incorporated into the overall judgement.
>
> **Weakness 3**
>
> > The experimental focus seems misplaced.
>
> We appreciate this comment. The goal of this work is to provide an effective approach for mobile agent evaluation. We propose TRACE to enhance the accuracy and automation of such evaluation. We also provide TRACEBench to systematically depict mobile agent capability. Like previous studies such as AndroidWorld, SPA-Bench and A3, we present the evaluation results of various mobile agents as the main result in the `Results section`. We hope revealing the performance of current mobile agents in practical deployment as well as their potential limitations can provide valuable insights for the optimization of mobile agents.
>
> As for the justification of TRACE’s effectiveness, we admit there is a lack of ablation study. We have supplemented ablation study result in the response to **Weakness 1**. We will add such contents to the revised manuscript.

---

> ### Author Response · Authors · 2025-11-25
>
> **Weakness 4**
>
>
> > The paper missed discussion with previous works on evaluting GUI agents in the related work section.
>
> We appreciate the reviewer’s constructive feedback. Existing online mobile agent evaluator can be divided into rule-based and model-based approaches. In the revised manuscript, we will enhance the discussion on existing mobile agent evaluator in the related work section.
>
> Here, we present comparison experiment results about model-based evaluator. Beside SPA-Bench, We run another method A3 (Android Agent Arena) in the proposed two settings: final state and essential state with both Qwen and GPT4o. *A3 final state* is almost same as *Final screen* schema shown in above table. *A3 essential state* first divides the given task into essential states like TRACE's milestone generation, then it applies a sliding window to the entire sequence to check whether each essential state is completed. This schema suffer from repeated computation and need to select a properate window length. The full results of comparison are as shown in `Table R2`. Compared with the human-annotated ground-truth, TRACE consistently achieves the highest accuracy and F1 score, which verifies that the results in Table 1 are highly reliable.
>
>
> | Method                     |   Accuracy |   Precision |   Recall |   F1-score |
> |:---------------------------|-----------:|------------:|---------:|-----------:|
> | TRACE                      |      **0.855** |      0.741 |    0.958 |      **0.836** |
> | SPA-Bench (Qwen)            |      0.532 |       0.453 |    **1.000** |      0.623 |
> | SPA-Bench (GPT4o)           |      0.726 |       0.600 |    0.875 |      0.712 |
> | A3 final state (Qwen)      |      0.823 |       0.810 |    0.708 |      0.756 |
> | A3 final state (GPT4o)     |      0.790 |       **0.824** |    0.583 |      0.682 |
> | A3 essential state (Qwen) |      0.705 |       0.625 |    0.625 |      0.625 |
> | A3 essential state (GPT4o)|      0.672 |       0.563 |    0.750 |      0.643 |
>
> *Table R5: Evaluation performance of TRACE, SPA-Bench, and A3 on 62 tasks sampled from TRACEBench.*
>
>
> Compared with rule-based evaluators, TRACE has better generality, automation, and stronger extensibility for new tasks. We evaluate trajectories of M3A collected on the widely used AndroidWorld benchmark (rule-based evaluator) with TRACE (Qwen2.5-VL-72B-Instruct as VLM evaluator). The evaluation results shown in below table indicate TRACE achieves good consistency (0.816 F1-score) to rule-based evaluator (AndroidWorld). Furthermore, we additionally compare TRACE with recent competitors such as SPA-Bench and A3. The results consistently show that TRACE outperforms these baselines.
>
> | Method | Accuracy | Precision | Recall | F1-score |
> |-------|-------|-------|-------|-------|
> | TRACE  | 0.921  | 0.833  |  0.800  | 0.816 |
> | SPA-Bench (Qwen) |  0.746  | 0.462  |  0.960 |  0.623 |
> | SPA-Bench (GPT4o) |   0.877 |   0.677 |    0.840 |  0.750 |
> |   A3 final state (Qwen) | 0.912  | 0.895  |  0.680 | 0.773|
> |   A3 final state (GPT4o) | 0.877 |  0.824  |  0.560  |0.667|
> |   A3 essential states (Qwen) | 0.851 |  0.682   | 0.600 | 0.638|
> |  A3 essential states (GPT4o) | 0.860  | 0.765   | 0.520 | 0.619|
>
> *Table R2: Evaluation performance of TRACE, SPA-Bench, and A3 on M3A trajectories collected on AndroidWorld.*

---

> ### Author Response · Authors · 2025-11-25
>
> **Question 1**
>
> > How does TRACE handle such dependencies when evaluating each step independently?
>
>
> The step-wise assessment only identify the effect of each executed action and extracts visual clues which are shown on the screenshot. This ensures that necessary evidence is properly collected. Then, during the overall judgment, TRACE selects the relevant criteria, based on the VLM's common sence and the generated task milestones, from the step-level evidence to determine whether the task is completed.
>
> For the case raised by the reviewer, although some actions may be unrelated to the task goal, for example, a “back" action taken for error correction, we still collect the effects of these actions during the step-wise assessment, such as “return to the previous page." This information is provided to the evaluator during the overall judgment, allowing it to make a decision based on all step-level evidence.
>
>
> **Question 2**
>
> > How does TRACE automatically generate task milestones? How does TRACE ensure that alternative valid solutions are not penalized?
>
>
> We leverage the VLM and prompts to generate milestones automatically.  The prompt used for milestone generation can be found in Appendix B.2.1.
>
> Regarding the issue of multiple valid ways for milestones and tasks, we have incorporated specific prompt designs to reduce the penalization of alternative valid solutions. Specifically, we added the requirement “Extract only the milestones necessary for judging task execution status" when instructing the VLM to generate milestones. This aims to prevent milestones from being overly tied to a particular execution strategy. From our perspective, any milestone that is strongly coupled with a specific execution pathway cannot be considered "necessary". Our milestone design follows a result-oriented checkpointing principle, rather than an execution-process-oriented one. For instance, in a simple task such as checking the weather, our prompt is designed so that milestones include only sub-goals necessary for verifying task success, such as “staying on the weather display page", and do not include specific execution trajectories like “opening a particular search engine or app."
>
> In addition, we also conducted ablation experiments to analyze the role of milestones in the judgment process. We found that the influence of milestones varies across different backend VLMs. As shown in Table R1, for Qwen, incorporating milestone information leads to a slightly positive performance gain, whereas for Doubao the effect is negative. However, the overall performance fluctuation remains small for both models. We recommend that when a stronger, closed-source VLM is used as the evaluator, providing milestones is unnecessary; however, when using a relatively weaker VLM, such as the open-source Qwen, incorporating milestones can improve evaluation accuracy. Overall, providing milestones to relatively weaker VLMs can, during the judgment process, help prevent them from overlooking essential steps and thereby reduce the likelihood of incorrectly categorizing failed tasks as successful.

---

> ### Author Response · Authors · 2025-11-25
>
> **Question 3**
>
> > Why did the authors not use existing benchmarks such as AiTW to demonstrate TRACE’s effectiveness and generality?
>
>
> TRACEBench's distinguishing features are that it closely reflects real-world usage scenarios and considers safety issues like unauthorized orders or payments when designing tasks. We agree that there are several good work on mobile agent benchmarks like mobile-bench serial and SPA-Bench, which provide lots of tasks in English and Chinese. Benefit from the scalablity of TRACEBench, tasks of existing benchmarks can be easily absorbed into TRACEBench, just like we adapt tasks from SPA-Bench. Different from the aforementioned work, TRACEBench contains 17 tasks related to placing order and so on, which touches the safety issue of mobile agent alongside evaluating helpfulness.
>
> As for AiTW, it is proposed to do offline evluation for mobile agent by action matching. The tasks of AiTW can indeed be used for evaluation, but AiTW cannot provide the ground truth required for online evaluation.
>
> Based on the reviewers' suggestions, we use existing benchmark AndroidWorld to demonstrate TRACE’s effectiveness and generality. As shown in above Table R2, TRACE achieves high consistency (0.816 F1-score) to AndroidWorld's rule-based evaluation results, which outperforms SPA-Bench with both Qwen and GPT4o.
>
>
> **Question 4**
>
> > Why are timeout and safety issues explicitly analyzed in the experimental section?
>
> The goal of this work is to provide an effective approach for mobile agent evaluation. The computational consumption and safety issue, as well as the task successful ratio, are different evaluation dimension of TRACEBench to depict the performance of mobile agents. The computational consumption is a common metric across previous work which reflects the efficiency and deployability on real devices. The safety metric captures the risk of undesired or unsafe behaviors during task execution. Together, they provide a more comprehensive and realistic evaluation of mobile agents and provide valuable insights for the optimization of mobile agents.
>
>
> **Question 5**
>
> > The experimental description for Table 1 is somewhat unclear. Are recall and precision defined by comparing the agent’s self-reported evaluation results with TRACE’s evaluation, or between TRACE’s evaluation and the human-annotated ground truth? If it is the former, how does this result demonstrate the effectiveness of TRACE as an evaluator?
>
> Table 1 does not include the metrics of recall and precision. If you mean CP and CR, these two metirc are introduced to reflect agent's awareness of task completion. These two metrics depict the accuracy of the agent’s own judgment regarding task completion with TRACE’s evaluation. If you are referring recall and precision in Table 2 or Table 3, in the first paragraph of `Section 5.2`, we have stated that TRACE’s evaluation results are compared with the human-annotated ground truth to validate the correctness and reliability of TRACE’s assessments.

---

> ### Comment · Reviewer_m47q · 2025-11-25
>
> Thank you for the clarifications. I still have some further questions.
>
> 1. **Research focus**
>
> > First, the core contribution of our work is the proposal of a new VLM-based evaluator, TRACE. Such an accurate and automated evaluator enables large-scale assessment of mobile agents and identify their respective limitations. The second contribution is the proposed TRACEBench consisting of 187 common tasks and the metric system. TRACEBench focuses on high-frequency tasks in everyday apps. Its distinguishing features are that it closely reflects real-world usage scenarios and considers safety issues like unauthorized orders or payments when designing tasks. The third contribution lies in key insights gained from evaluating eight representative mobile agents with TRACE on TRACEBench.
>
> The authors claim that the main contribution of this paper is the algorithmic contribution of the new design of the evaluator, with other two contributions on the new benchmarks and evaluations of current methods. However, the primary area of this paper is *datasets and benchmarks*, making the position of this paper unclear. Meanwhile, as raised by other reviewers, I think the contribution of TRACEBench is quite unclear. What is the advantage of using TRACEBench compared with other latested benchmarks.
>
> 2. **Ablation study on TRACE**
>
> Thank you for providing a comprehensive ablation study. There are still some questions regarding the ablation study. 1) Why using only last screenshot performs better then using trajectory with Qwen models? I assume that trajectory will always contain the last screenshot. 2) The ablation study got two totally different conclusions with two different models, making the method and conclusion in the paper less convincing. It looks like that simply using a better model is more effective than using TRACE instead of using a normal evaluation method. Also, the ablation study needs to test on more base models including those sota models like gemini 3.0 or gpt5.1, since the two results are not converged.
>
> 3. **Writing issues**
>
> Thank the authors to clarify the questions about the paper writing. I think the paper need a thoroughly rewritten to make it more clear based on the current version. Also for the flow of the paper, which makes the focus of the paper unclear, needs to be carefully redesigned. Maybe the comparison of time and safety should be moved to appendix.

---

> > ### Author Response · Authors · 2025-11-27
> >
> > We sincerely thank the reviewer for the detailed follow-up comments.
> >
> > **1. Research focus**
> >
> > According to the ICLR 2026 official subject areas, we think *datasets and benchmarks* is the most appropriate and relevant category for our submission, since our paper proposes a new VLM-based evaluation approach TRACE. We also construct TRACEBench and evaluate 8 representative mobile agents with TRACE. If the reviewer disagree with that, we kindly request guidance on which subject area would be more appropriate for this work.
> >
> > As for the contribution of TRACEBench, it lies in the following aspects: (1) Real-world relevance: TRACEBench is built entirely from common tasks in everyday mobile apps, including safety-critical scenarios such as ordering and payments, making it a benchmark truly aligned with practical deployment. (2) Comprehensive metrics: It provides a richer evaluation system covering execution ability (SR, OTR), completion awareness (CR, CP), safety (SFR), and computational cost (TEX). (3) Scalability: TRACEBench requires no manual annotations and can be easily expanded with new tasks from existing benchmarks.
> >
> > **2. Ablation study on TRACE**
> >
> > > 1) Why using only last screenshot performs better then using trajectory with Qwen models? I assume that trajectory will always contain the last screenshot.
> >
> > We found in our *Final Screen* analysis that Qwen exhibits stronger hallucination tendencies than Doubao. For tasks that are actually completed but where the last screenshot alone does not provide sufficient evidence of completion, Qwen often hallucinates that the task has been completed and occasionally gets it right by chance, whereas Doubao relies strictly on the visible information in the final frame and labels such cases as failures when evidence is insufficient. Intuitively, *Final Screen* should perform worse than *Trajectory*, yet Qwen’s hallucination bias leads to unexpectedly higher scores.
> >
> > For TRACE, since we provide information of every step, the VLM does not need to hallucinate the execution process and can make judgments based on the actual information at each step, resulting in more accurate evaluations.
> >
> > >  2) The ablation study got two totally different conclusions with two different models, making the method and conclusion in the paper less convincing. It looks like that simply using a better model is more effective than using TRACE instead of using a normal evaluation method.
> >
> > We don't know which conclusion is totally different. We have explained the effect of milestone and clarified why *Final Screen (Qwen)* gets a relative high score. As for "using a better model is more effective than using TRACE"，we cannot agree because *TRACE* consistently outperforms *Trajectory* across Qwen and doubao. The comparison results between TRACE and other pervious VLM-based evaluators on both TRACEBench and AndroidWorld also prove the effectiveness of TRACE.
> >
> > > the ablation study needs to test on more base models including those sota models like gemini 3.0 or gpt5.1
> >
> > We appreciate the reviewer’s suggestion to test TRACE on more base models, including SOTA models such as Gemini 3.0 or GPT-5.1 (released in mid-November). However, we believe that performing ablation on these very recent models is unlikely to yield fundamentally new insights. Moreover, one of the key advantages of TRACE is that it can achieve reliable and meaningful evaluation even with relatively weaker or cheaper base models.
> >
> >
> > **3. Writing issues**
> >
> > Thanks for your valuable suggestions. We will upload a revised manuscript as soon as possible.

---

### Official Review · Reviewer_1ef2 · 2025-10-31

**Soundness:** 2
**Presentation:** 2
**Contribution:** 2
**Rating:** 2
**Confidence:** 3

**Summary:**

This paper introduces TRACE (TRajectory-based Automated Coarse-to-fine Evaluation), a fully automated VLM-based framework for evaluating mobile GUI agents. TRACE adopts a two-stage coarse-to-fine design that first analyzes individual actions and then performs an overall judgment based on aggregated evidence. The authors also release TRACEBench, a benchmark of 187 mobile tasks across 35 apps with explicit safety-focused cases. Experiments show that TRACE achieves an F1 score of 0.836 against human annotations, outperforming SPA-Bench and demonstrating improved reliability and automation. The work aims to advance realistic, safety-aware, and scalable evaluation of mobile agents.

**Strengths:**

1. **Approach and design**: TRACE proposes a coarse-to-fine decomposition that isolates local reasoning from global evaluation, substantially reducing VLM context complexity. This staged design is intuitive and effective, producing a strong correlation (F1 = 0.836) with human judgment while remaining fully automated.
2. **Practical and accessible implementation**: TRACE attains competitive evaluation accuracy using an open-source VLM (Qwen2.5-VL-72B-Instruct). The reported improvements in F1 score, precision, and recall over SPA-Bench provide solid empirical evidence of its effectiveness.
3. **Clarity and presentation quality**:The paper is clearly written and well structured, with sufficient methodological descriptions and illustrative examples that make the proposed framework easy to follow. The inclusion of multiple VLM evaluators (Qwen, Doubao, GPT-4o, Gemini) also helps demonstrate the robustness and general applicability of the proposed approach.

**Weaknesses:**

1. **Unfocused contribution scope**: The paper claims three contributions—pipeline, benchmark, and safety detection—but only the pipeline represents a substantive methodological novelty. The benchmark and safety modules, though useful, blur the central narrative. A tighter focus on evaluating the pipeline across existing benchmarks (e.g., Mobile-Bench, SPA-Bench) would better clarify its impact.
2. **Limited benchmark novelty and scale**: TRACEBench’s 187 tasks are modest compared with prior work (e.g., Mobile-Bench 832, Mobile-Bench-v2 > 12 k). The Chinese-only design restricts accessibility and limits cross-lingual generality. The rationale for this language choice and the proportion of newly designed versus adapted tasks should be better justified.
3. **Insufficient safety evaluation rigor**: The safety detector is prompt-based and lacks quantitative validation (precision/recall) or baseline comparison against simple heuristics. Without such analysis, it appears closer to prompt engineering than a verified safety metric. The trade-off between Success Ratio (SR) and Safety Ratio (SFR) could also be explored more systematically.
4. **Lack of ablations and pipeline analysis**: Although the step-wise design is conceptually strong, the paper does not test simpler baselines such as (i) final-screenshot-only or (ii) single-pass full-trajectory evaluation. An ablation quantifying how much each stage contributes to accuracy would strengthen the argument that coarse-to-fine evaluation is necessary.
5. **Uncontrolled execution environment and variance**: TRACEBench relies on real mobile devices, introducing nondeterminism (ads, residual states). The paper acknowledges this but does not quantify benchmark variance or report results from repeated runs. Without this, the statistical significance of agent performance differences remains unclear.
6. **Limited comparison to other VLM/LLM evaluators**: SPA-Bench is the only baseline. Other recent works (e.g., Android Agent Arena [1] and OSUniverse [2]) also propose automated evaluators using VLMs or LLMs. Evaluating against these would clarify TRACE’s relative advantage.
7. **Milestone reliability not evaluated**: The overall judgment phase critically depends on accurate milestone decomposition. Errors in milestone extraction could cascade into incorrect final judgments, yet no quantitative assessment of this stage’s reliability is provided.
8. **Metric definition and motivation**: Metrics such as Complete Recall/Precision (CR/CP) are introduced but not well-motivated relative to standard success metrics. Concrete examples showing when CR/CP provide distinct insights would help justify their inclusion.

### **References**
[1] Android Agent Arena: A3 Benchmark for Mobile GUI Agents. https://arxiv.org/abs/2501.01149

[2] OSUniverse: Benchmark for Multimodal GUI-Navigation AI Agents. https://arxiv.org/abs/2505.03570

**Questions:**

1. **Benchmark motivation**: What unique features of TRACEBench justify introducing a new dataset instead of expanding existing ones such as Mobile-Bench or SPA-Bench? How many of the 187 tasks are genuinely new versus adapted?
2. **Language choice**: Why were all task instructions written in Chinese, including for global apps?
3. **Environment control and reproducibility**: When using real devices, how do you ensure fairness across runs? Was any emulator, snapshotting, or re-initialization used?
4. **Milestone reliability**: Have you measured the correctness of automatically generated milestones? How sensitive is overall performance to errors at this stage?
5. **Metric usefulness**: Please illustrate cases where CR/CP provide insights beyond success ratio, or consider simplifying the metrics if redundant.

---

> ### Author Response · Authors · 2025-11-25
>
> Thank you for your time and valuable feedback.
>
> ### Q1. Unfocused contribution scope.
>
> We are sorry that the current paper presentation gave the impression of "unfocused contribution scope".
>
> From our perspective, the core contribution of this paper is the proposal of a new VLM-based evaluator called TRACE which is more accurate, general, and fully automated. We design two experiments to demonstrate TRACE's effectiveness and robustness. We acknowledge that the paper lacks a more detailed ablation study for TRACE, and we will present the corresponding ablation results and detailed analysis below.
>
> The second contribution is the proposed TRACEBench consisting of 187 common tasks and the metric system. TRACEBench focuses on high-frequency tasks in everyday apps. Its distinguishing features are that it closely reflects real-world usage scenarios and considers safety issues like unauthorized orders or payments when designing tasks.
>
> The third contribution lies in key insights gained from evaluating eight representative mobile agents with TRACE on TRACEBench.
>
> ### Q2. Benchmark motivation and language choice.
>
> As clarified before, TRACEBench's distinguishing features are that it closely reflects real-world usage scenarios and considers safety issues like unauthorized orders or payments when designing tasks. We agree that there are several good work on mobile agent benchmarks like mobile-bench serial and SPA-Bench, which provide lots of tasks in English and Chinese. Benefit from the scalablity of TRACEBench, tasks of existing benchmarks can be easily absorbed into TRACEBench, just like we adapt tasks from SPA-Bench. Different from the aforementioned work, TRACEBench contains 17 tasks related to placing order and so on, which touches the safety issue of mobile agent alongside evaluating helpfulness. Besides, the metric system of TRACEBench covers agents' task execution ability (SR, OTR), awareness of task completion (CR, CP), safety (SFR), and computational consumption (TEX).
>
> Furthermore, to demonstrate the language robustness of our method TRACE, we evaluate trajectories of M3A collected on the widely used AndroidWorld benchmark (116 English tasks with English system environment) with TRACE (Qwen2.5-VL-72B-Instruct as VLM evaluator). The evaluation result achieve 0.816 F1-score to the ground-truth (AndroidWorld's rule-based evaluator), showing that TRACE remains effective across languages. We additionally compare TRACE with recent competitors such as SPA-Bench and A3. The results consistently show that TRACE outperforms these baselines.
>
> | Method | Accuracy | Precision | Recall | F1-score |
> |-------|-------|-------|-------|-------|
> | TRACE  | 0.921  | 0.833  |  0.800  | 0.816 |
> | SPA-Bench (Qwen) |  0.746  | 0.462  |  0.960 |  0.623 |
> | SPA-Bench (GPT4o) |   0.877 |   0.677 |    0.840 |  0.750 |
> |   A3 final state (Qwen) | 0.912  | 0.895  |  0.680 | 0.773|
> |   A3 final state (GPT4o) | 0.877 |  0.824  |  0.560  |0.667|
> |   A3 essential states (Qwen) | 0.851 |  0.682   | 0.600 | 0.638|
> |  A3 essential states (GPT4o) | 0.860  | 0.765   | 0.520 | 0.619|
>
> *Table R2: Evaluation performance of TRACE, SPA-Bench, and A3 on M3A trajectories collected on AndroidWorld.*
>
> ### Q3: Insufficient safety evaluation rigor.
>
> Thank you for raising this question. Based on the reviewers' suggestions, we ask an expert to annotate the operation safety of 17 safety-related trajectories of UI-TARS as ground-truth. The quantitative validation results of TRACE's safety detector are as follow:
>
> | Method | Accuracy | Precision | Recall | F1-score |
> |-------|-------|-------|-------|-------|
> | TRACE  | 0.882  | 0.778  |  1.000  | 0.875 |
>
> *Table R4: Quantitative validation of TRACE safty detector with 17 safety-related tasks.*
>
>
> Besides, Table 1 in the submitted manuscript clearly illustrates the inherent trade-off between the Success Ratio (SR) and the Safety Ratio (SFR)—agents with higher SR tend to show lower SFR, and vice versa. We acknowledge that the safety evaluation of TRACE is still relatively preliminary, and we plan to design more in-depth safety evaluations in future work.

---

> ### Author Response · Authors · 2025-11-25
>
> ### Q4: More ablations and Milestone reliability.
>
> For the ablation study, we ablate TRACE into four variants: (i) *TRACE without milestone* (no milestone information is provided in TRACE overall judgement), (ii) *Final screen* (only using final screen for judgement), (iii) *Trajectory* (all screenshots, actions, and milestone are provided in a single query), and (iv) *Trajectory without milestone* (all screenshots and actions are provided in a single query). Each variant adopts minimal modifications on prompt template to maintain maximal similarity to TRACE. We evaluate these variants with both Qwen2.5-VL-72B-Instruct and doubao-1.5-vision-pro-250328, and the results are shown in below table. Compared with *Final screen* and *Trajectory*, TRACE shows superior evaluation performance with both two VLMs, which demonstrates the effectiveness of TRACE's design decisions. *Final screen* schema doesn't provide enough information for trajectory evaluation. *Trajectory* schema inevitably induces very long and complex contexts, which makes it difficult for VLMs to accuratly understand and evaluate the trajectory. This schema is also not suitable for very long trajectory due to the context window limitaion. As for the milestone information used in TRACE, its impact depends on the chosen VLM evaluator. For Qwen, incorporating milestone information leads to a slightly positive performance gain, whereas for Doubao the effect is negative. However, the overall performance fluctuation remains small for both models. We recommend that when a stronger, closed-source VLM is used as the evaluator, providing milestones is unnecessary; however, when using a relatively weaker VLM, such as the open-source Qwen, incorporating milestones can improve evaluation accuracy.
>
> | Evaluator | VLM | Accuracy | Precision | Recall | F1-score |
> |:---------|:---------|---------:|---------:|---------:|---------:|
> | TRACE  | Qwen2.5-VL-72B-Instruct  | **0.855**  |  **0.741**  |  **0.958**  | **0.836**  |
> | TRACE w/o milestone  | Qwen2.5-VL-72B-Instruct  | 0.823  |  0.710  |  0.917  |  0.800  |
> | Final screen | Qwen2.5-VL-72B-Instruct  |  0.774  |  0.667  |  0.833  |  0.741  |
> | Trajectory | Qwen2.5-VL-72B-Instruct  | 0.613  |  0.500  |  **0.958** |  0.657  |
> | Trajectory w/o milestone | Qwen2.5-VL-72B-Instruct  |  0.565  |  0.469  |  **0.958**  |  0.630  |
> | TRACE  | doubao-1.5-vision-pro-250328  | 0.903  |  0.909  |  0.833  | 0.870  |
> | TRACE w/o milestone  | doubao-1.5-vision-pro-250328  | **0.935**  |  **0.955**  |  **0.875**  |  **0.913**  |
> | Final screen | doubao-1.5-vision-pro-250328  |  0.710  |  0.688  |  0.458  |  0.550  |
> | Trajectory | doubao-1.5-vision-pro-250328  | 0.790  |  0.720  |  0.750  |  0.734  |
> | Trajectory w/o milestone | doubao-1.5-vision-pro-250328  |  0.806  |  0.773  |  0.708  |  0.739  |
>
> *Table R1: Ablation study of TRACE.*
>
> ### Q5: Environment control and reproducibility.
>
> To assess the variance introduced by real-device execution, we conduct additional experiments using three different mobile phones on the subset of TRACEBench. We ran M3A and evaluated the collected trajectories with TRACE, focusing on Success Ratio (SR). As we can see, the evaluation results exhibit good consistency, with only minimal standard deviation 0.024 observed in SR across runs. This result indicates that TRACEBench remains stable across heterogeneous real-device environments and that the performance differences reported in the paper are statistically reliable.
>
> |        | Trial 1   | Trial 2 | Trial 3 | Mean | std |
> |-------|---------|------|-------|---------|------|
> | SR     | 0.269 | 0.212  | 0.250  | 0.244  | 0.024 |
>
> *Table R3: Run M3A agent with three different android device and then evaluate with TRACE.*

---

> ### Author Response · Authors · 2025-11-25
>
> ### Q6: More comparison to other VLM/LLM evaluators.
>
> Thank you for sharing these works. We run Android Agent Arena (A3) in the proposed two settings: final state and essential state with both Qwen and GPT4o. *A3 final state* is almost same as *Final screen* schema shown in above table. *A3 essential state* first divides the given task into essential states like TRACE's milestone generation, then it applies a sliding window to the entire sequence to check whether each essential state is completed. This schema suffer from repeated computation and need to select a properate window length. As shown in below table, TRACE outperforms A3 in both settings with both Qwen and GPT4o. The result of *A3 final state* also shows good consistency to the *Final screen* shown above.
>
> | Method                     |   Accuracy |   Precision |   Recall |   F1-score |
> |:---------------------------|-----------:|------------:|---------:|-----------:|
> | TRACE                      |      **0.855** |      0.741 |    0.958 |      **0.836** |
> | SPA-Bench (Qwen)            |      0.532 |       0.453 |    **1.000** |      0.623 |
> | SPA-Bench (GPT4o)           |      0.726 |       0.600 |    0.875 |      0.712 |
> | A3 final state (Qwen)      |      0.823 |       0.810 |    0.708 |      0.756 |
> | A3 final state (GPT4o)     |      0.790 |       **0.824** |    0.583 |      0.682 |
> | A3 essential state (Qwen) |      0.705 |       0.625 |    0.625 |      0.625 |
> | A3 essential state (GPT4o)|      0.672 |       0.563 |    0.750 |      0.643 |
>
> *Table R5: Evaluation performance of TRACE, SPA-Bench, and A3 on 62 tasks sampled from TRACEBench.*
>
> ### Q7: Illustrate cases where CR/CP provide insights beyond success ratio.
>
> CR and CP are introduced to reflect agent's awareness of task completion. These two metrics depict the accuracy of the agent’s own judgment regarding task completion. In contrast, SR evaluates task success solely based on the execution trajectory and is not affected by the agent’s self-reported completion status.

---

> > ### Comment · Reviewer_1ef2 · 2025-11-27
> > **Response to Authors' Rebuttal**
> >
> > Thanks for the response and for addressing part of my concerns. I agree that in terms of evaluation, TRACE demonstrates stronger performance compared with existing methods, and the additional experiments help make the method itself more convincing. That said, I share the concern raised by reviewer m47q. As a dataset and benchmark paper, I feel the contribution and novelty are still not sufficient on the dataset side or for the overall benchmark framework, even after going through the rebuttal and the other reviews. I also think the paper would require substantial refinement and rewriting to meet the bar for ICLR. I will adjust my rating and raise the score, though this will be my final decision.

---

### Official Review · Reviewer_Ae6i · 2025-10-31

**Soundness:** 3
**Presentation:** 3
**Contribution:** 3
**Rating:** 6
**Confidence:** 4

**Summary:**

This paper addresses the challenges of evaluating mobile agents in real-world environments by proposing TRACE, a fully automated, VLM-based "coarse-to-fine" evaluation framework. TRACE performs step-wise assessment to extract screen clues, action effects, and safety risks, followed by an overall judgment stage that integrates task milestones and the final state. The authors also construct TRACEBench, a benchmark comprising 187 tasks across 35 commonly used applications, with a particular emphasis on safety and real-world scenarios. Extensive experiments on eight representative mobile agents demonstrate the advantages of TRACE in terms of accuracy, generalizability, and automation.

**Strengths:**

1.TRACE decomposes evaluation into step-level analysis and overall judgment, significantly reducing the complexity of understanding long trajectories and improving assessment accuracy and reliability.
2.RACEBench covers diverse daily applications and tasks and deliberately includes 17 risky tasks, systematically introducing quantitative safety metrics for mobile agent evaluation for the first time.
3.TRACE operates without manual annotations or predefined rules, relying solely on task descriptions, screenshots, and action sequences. It supports evaluation across environments, tasks, and agent architectures, demonstrating strong scalability.

**Weaknesses:**

1. Running TRACEBench directly on real devices means environmental noise (e.g., pop-up ads, residual states) can lead to inconsistent results. While parameterization and repeated runs are proposed as mitigations, this may still affect evaluation stability.
2.Experiments show TRACE's performance varies with the chosen VLM (e.g., Doubao performs best, while GPT-4o and Gemini struggle on some tasks). This could limit its applicability in resource-constrained or specific linguistic environments.
3.Although TEX and TEV metrics are mentioned, there is insufficient analysis of practical deployment concerns like computational overhead and response time for large-scale evaluations.

**Questions:**

1.	Beyond parameterization and repeated runs, does TRACE incorporate other mechanisms (e.g., context-aware reasoning or dynamic adaptation) to enhance robustness against environmental stochasticity like ads or dynamic content?
2.	Have you considered adapting TRACE for non-Chinese environments or other mobile platforms (e.g., iOS)? Would your recommendations for VLM selection differ in multi-lingual or cross-platform scenarios?

---

> ### Author Response · Authors · 2025-11-25
>
> We thank the reviewer for valuable feedback. Below we provide point-by-point responses and clarifications.
>
> **Weakness 1**
> > Running TRACEBench directly on real devices can lead to inconsistent results.
>
> We agree with the reviewer that real devices inevitably introduce environmental noise, which may affect the reproducibility of evaluations. However, we believe that assessing mobile agent performance in real environments is an essential step before real-world deployment. Existing studies, such as AndroidWorld, conduct evaluations in closed Android emulator environments, which cannot fully reflect an agent’s performance in real-world scenarios, for example the robustness to pop-up ads. Moreover, some apps, such as Meituan and Taobao, cannot run properly on emulators due to anti-crawling mechanisms, making real-device evaluation necessary. We supplement an additional experiment in which we evaluate M3A agent with three different android devices (repeated runs) and the result is shown below. On the same subset of TRACEBench, the std of three trials is 0.024, which means good evaluation consistency.
>
> |        | Trial 1   | Trial 2 | Trial 3 | Mean | std |
> |-------|---------|------|-------|---------|------|
> | SR     | 0.269 | 0.212  | 0.250  | 0.244  | 0.024  |
>
> *Table R3: Run M3A agent with three different android device and then evaluate with TRACE.*
>
>
>
> **Weakness 2**
> > The selection of VLM
>
> Indeed, Table 3 in the submitted manuscript shows that GPT-4o and Gemini achieve lower F1 scores compared with Qwen and Doubao. However, a closer analysis shows that this gap primarily appears in recall, while in terms of precision, GPT-4o and Gemini perform on par with or even better than Qwen and Doubao. This indicates that under the current prompt, GPT-4o and Gemini apply a stricter evaluation criterion. If users wish to substitute to other VLM evaluators like GPT-4o or Gemini, they can adapt the existing prompt by slightly relaxing the overall judgment to ensure compatibility.
>
>
> **Weakness 3**
> > Insufficient analysis of practical deployment concerns.
>
> For compuration-related metric, we report token consumption rather than time-based metrics. This choice avoids the influence of external factors such as network fluctuations, leading to a fairer comparison across methods. TEX is a metric of TRACEBench which is used to evaluate the computational consumption of agent execution. TEV is designed to depict the computational consumption of evaluator. For example, as shown in Table 2, the computational overhead of SPA-Bench and TRACE is similar when using Qwen as VLM evaluator.

---

> ### Author Response · Authors · 2025-11-25
>
> **Question 1**
> > Does TRACE incorporate other mechanisms to enhance robustness against environmental stochasticity?
>
> We think the design of TRACE inherently induces the robustness against environmental stochasticity. The step-wise assessment not only focuses on the effect of each executed action but also extracts visual clues which are not related to current action but revelent to the given task. This ensures that necessary evidence is properly collected. Then, during the overall judgment, TRACE selects the relevant criteria, based on the VLM's common sence and the generated task milestones, from the step-level evidence to determine whether the task is completed. This design makes the evaluation results robust to environmental stochasticity.
>
> **Question 2**
> > Have you considered adapting TRACE for non-Chinese environments or other mobile platforms (e.g., iOS)? Would your recommendations for VLM selection differ in multi-lingual or cross-platform scenarios?
>
> We have not yet adapted TRACE to other platforms such as iOS or the web. However, we believe the underlying concept of TRACE is not limited to mobile devices. Extending TRACE to cross-platform scenarios will be part of our future work.
>
> To demonstrate the language robustness of our method TRACE, we evaluate trajectories of M3A collected on the widely used AndroidWorld benchmark (116 English tasks with English system environment) with TRACE (Qwen2.5-VL-72B-Instruct as VLM evaluator). The evaluation result achieve 0.816 F1-score to the ground-truth (AndroidWorld's rule-based evaluator), showing that TRACE remains effective across languages. We additionally compare TRACE with recent competitors such as SPA-Bench and A3. The results consistently show that TRACE outperforms these baselines.
>
> | Method | Accuracy | Precision | Recall | F1-score |
> |-------|-------|-------|-------|-------|
> | TRACE  | 0.921  | 0.833  |  0.800  | 0.816 |
> | SPA-Bench (Qwen) |  0.746  | 0.462  |  0.960 |  0.623 |
> | SPA-Bench (GPT4o) |   0.877 |   0.677 |    0.840 |  0.750 |
> |   A3 final state (Qwen) | 0.912  | 0.895  |  0.680 | 0.773|
> |   A3 final state (GPT4o) | 0.877 |  0.824  |  0.560  |0.667|
> |   A3 essential states (Qwen) | 0.851 |  0.682   | 0.600 | 0.638|
> |  A3 essential states (GPT4o) | 0.860  | 0.765   | 0.520 | 0.619|
>
> *Table R2: Evaluation performance of TRACE, SPA-Bench, and A3 on M3A trajectories collected on AndroidWorld.*
>
> As for the VLM selection, for testing Chinese apps, we recommend using Doubao. For global apps, we suggest GPT-4o or Gemini, but with a slightly relaxed overall-judgment prompt.

---

### Official Review · Reviewer_hJvM · 2025-10-31

**Soundness:** 2
**Presentation:** 1
**Contribution:** 2
**Rating:** 4
**Confidence:** 3

**Summary:**

The paper proposes TRACE, a coarse-to-fine method for judging mobile agent trajectories, along with a new benchmark for mobile agents, TRACEBench. The paper evaluates and analyzes several existing models on TRACEBench.

**Strengths:**

* The paper proposes a new benchmark for mobile agents, with greater focus on apps popular in China and on safety considerations.
* The paper proposes a new methodology for building LLM evaluators for mobile agents, with improvements in F1 over a method from prior work.
* The paper analyzes the performance of several agents on the proposed benchmark.

**Weaknesses:**

* The paper has several contributions: (1) TRACE, a new method for building a LLM evaluator by decomposing the task into a pipeline system, (2) TRACEBench, a new benchmark for mobile agents, and (3) empirical analysis of several models on TRACEBench. However, each contribution is a bit underdeveloped. For example, the paper lacks justification for TRACE's complexity (see point below), despite this contribution being most emphasized in the title and narrative. The new benchmark could potentially be a more compelling contribution, but it is not well justified with respect to existing benchmarks like AndroidWorld. A new benchmark focusing on apps more popular in China could be useful to test i18n capabilities of agents beyond English, but I didn't quite understand the justification otherwise. A greater focus on safety judgements could also be useful, but this contribution was a bit lost in the current narrative.
* Prior work has also proposed various pipelined systems for improving automated UI agent evaluation, e.g. WebJudge from https://arxiv.org/abs/2504.01382. Since this paper proposes a relatively complex pipeline for evaluation, it would be useful to perform a more comprehensive analysis of which specific choices are most important and how they compare with prior work. For multimodal models with sufficient context windows, how does the approach compare with naively encoding the complete trajectory (the paper claims performance would be better, but this claim is not empirically supported)? Does this difference depend specifically on the context window and screen understanding capabilities of the underlying model? Without more analysis, it's difficult to understand the key take-aways from the various design decisions of TRACE. If a new LLM evaluator is indeed the key contribution, it would be useful to evaluate it across multiple benchmarks, e.g. you could analyze agreement with heuristic rewards on AndroidWorld compared with LLM evaluators from prior work.

Summary: I think the paper could be improved for a future submission by either (1) focusing on the benefits of the proposed benchmark, relaxing some of the claims related to the specific LLM evaluator or (2) expanding the empirical analysis of different design choices related to LLM evaluators, extending comparisons of different evaluators to existing benchmarks.

Nits:

* The method is described as "coarse-to-fine", but TRACE is actually the reverse. It starts with fine step-level judgements and then makes a coarse trajectory judgement. This was perhaps not the best terminology.
* Figure 1 - Why no arrow from (4) to (5)?

**Questions:**

See weaknesses above.

---

> ### Author Response · Authors · 2025-11-25
>
> We thank the reviewer for valuable feedback. Below we provide point-by-point responses and clarifications.
>
>
> **Weakness 1**
> > Each contribution is a bit underdeveloped.
>
> We are sorry that the current paper presentation gave the impression that "each contribution is a bit underdeveloped."
>
> From our perspective, the core contribution of this paper is the proposal of a new VLM-based evaluator called TRACE which is more accurate, general, and fully automated. We design two experiments to demonstrate TRACE's effectiveness and robustness. We acknowledge that the paper lacks a more detailed ablation study for TRACE, and we will present the corresponding ablation results and detailed analysis below.
>
> The second contribution is the proposed TRACEBench consisting of 187 common tasks and the metric system. TRACEBench focuses on high-frequency tasks in everyday apps. Its distinguishing features are that it closely reflects real-world usage scenarios and considers safety issues like unauthorized orders or payments when designing tasks.
>
> The third contribution lies in key insights gained from evaluating eight representative mobile agents with TRACE on TRACEBench.
>
> > Lack justification for TRACE's complexity (see point below).
>
> Apart from two experiments presented in the submitted manuscript, we supplement ablation study and the comparison with SPA-Bench and A3 (Andirod Agent Arena) on AndroidWorld. Please see our response to **Weakness 2** provided below.
>
> > The new benchmark is not well justified with respect to existing benchmarks like AndroidWorld.
>
> As clarified before, TRACEBench's distinguishing features are that it closely reflects real-world usage scenarios and considers safety issues like unauthorized orders or payments when designing tasks. The main differences lie in the types of apps covered and the types of tasks included. AndroidWorld provides 116 tasks across 20 Android apps, but these apps differ significantly from mainstream app designs. SPA-Bench provided 340 tasks covering 58 commonly used apps with three difficulty level. The three difficulty levels are merely extensions of the same task in terms of complexity. Moreover, each app includes at most three task types, which limits the coverage of real app usage scenarios and, consequently, cannot fully reflect an agent’s ability to operate a given app. Benefit from the scalablity of TRACEBench, tasks of existing benchmark can be easily absorbed into TRACEBench, just like we adapt tasks from SPA-Bench. Moreover, TRACEBench contains 17 tasks related to placing order and so on, which touches the safety issue of mobile agent. This aspect is not covered by existing general benchmarks.
>
> > A greater focus on safety judgements is a bit lost in the current narrative.
>
> We agree that our work only touch safety judgements and is not sufficiently in-depth. In this study, we focus only on risk scenarios such as placing orders and making payments. Since TRACEBench operates in real environments, it is challenging to cover a broader range of risky tasks. Existing benchmarks such as MobileSafetyBench and MLA-Trust specifically focus on safety, and we plan to design more in-depth safety evaluations in future work.

---

> ### Author Response · Authors · 2025-11-25
>
> **Weakness 2**
> > Without more analysis, it's difficult to understand the key take-aways from the various design decisions of TRACE.
>
> We provide a detailed ablation study to probe the design choices of TRACE. In addition, we include a new comparative experiment with SPA-Bench and A3 (Andirod Agent Arena) on AndroidWorld. The detailed analyses are presented below.
>
>
> (1) *TRACE ablation.* For the ablation study, we ablate TRACE into four variants: (i) *TRACE without milestone* (no milestone information is provided in TRACE overall judgement), (ii) *Final screen* (only using final screen for judgement), (iii) *Trajectory* (all screenshots, actions, and milestone are provided in a single query), and (iv) *Trajectory without milestone* (all screenshots and actions are provided in a single query). Each variant adopts minimal modifications on prompt template to maintain maximal similarity to TRACE. We evaluate these variants with both Qwen2.5-VL-72B-Instruct and doubao-1.5-vision-pro-250328, and the results are shown in below table. Compared with *Final screen* and *Trajectory*, TRACE shows superior evaluation performance with both two VLMs, which demonstrates the effectiveness of TRACE's design decisions. *Final screen* schema doesn't provide enough information for trajectory evaluation. *Trajectory* schema inevitably induces very long and complex contexts, which makes it difficult for VLMs to accuratly understand and evaluate the trajectory. This schema is also not suitable for very long trajectory due to the context window limitaion. As for the milestone information used in TRACE, its impact depends on the chosen VLM evaluator. For Qwen, incorporating milestone information leads to a slightly positive performance gain, whereas for Doubao the effect is negative. However, the overall performance fluctuation remains small for both models. We recommend that when a stronger, closed-source VLM is used as the evaluator, providing milestones is unnecessary; however, when using a relatively weaker VLM, such as the open-source Qwen, incorporating milestones can improve evaluation accuracy.
> | Evaluator | VLM | Accuracy | Precision | Recall | F1-score |
> |:---------|:---------|---------:|---------:|---------:|---------:|
> | TRACE  | Qwen2.5-VL-72B-Instruct  | **0.855**  |  **0.741**  |  **0.958**  | **0.836**  |
> | TRACE w/o milestone  | Qwen2.5-VL-72B-Instruct  | 0.823  |  0.710  |  0.917  |  0.800  |
> | Final screen | Qwen2.5-VL-72B-Instruct  |  0.774  |  0.667  |  0.833  |  0.741  |
> | Trajectory | Qwen2.5-VL-72B-Instruct  | 0.613  |  0.500  |  **0.958** |  0.657  |
> | Trajectory w/o milestone | Qwen2.5-VL-72B-Instruct  |  0.565  |  0.469  |  **0.958**  |  0.630  |
> | TRACE  | doubao-1.5-vision-pro-250328  | 0.903  |  0.909  |  0.833  | 0.870  |
> | TRACE w/o milestone  | doubao-1.5-vision-pro-250328  | **0.935**  |  **0.955**  |  **0.875**  |  **0.913**  |
> | Final screen | doubao-1.5-vision-pro-250328  |  0.710  |  0.688  |  0.458  |  0.550  |
> | Trajectory | doubao-1.5-vision-pro-250328  | 0.790  |  0.720  |  0.750  |  0.734  |
> | Trajectory w/o milestone | doubao-1.5-vision-pro-250328  |  0.806  |  0.773  |  0.708  |  0.739  |
>
> *Table R1: Ablation study of TRACE.*
>
> (2) *Comparison to other vlm-based evaluator on AndroidWorld.* Based on the reviewer's suggestion, we evaluate TRACE with AndroidWorld and analyze agreement with heuristic rewards compared with pervious evaluators including SPA-Bench and A3. As shown in Table R2, TRACE achieves high consistency (0.816 F1-score) to AndroidWorld's rule-based evaluation results, which outperforms SPA-Bench and A3 with both Qwen and GPT4o.
>
>
> | Method | Accuracy | Precision | Recall | F1-score |
> |-------|-------|-------|-------|-------|
> | TRACE  | 0.921  | 0.833  |  0.800  | 0.816 |
> | SPA-Bench (Qwen) |  0.746  | 0.462  |  0.960 |  0.623 |
> | SPA-Bench (GPT4o) |   0.877 |   0.677 |    0.840 |  0.750 |
> |   A3 final state (Qwen) | 0.912  | 0.895  |  0.680 | 0.773|
> |   A3 final state (GPT4o) | 0.877 |  0.824  |  0.560  |0.667|
> |   A3 essential states (Qwen) | 0.851 |  0.682   | 0.600 | 0.638|
> |  A3 essential states (GPT4o) | 0.860  | 0.765   | 0.520 | 0.619|
>
> *Table R2: Evaluation performance of TRACE, SPA-Bench, and A3 on M3A trajectories collected on AndroidWorld.*

---

> ### Author Response · Authors · 2025-11-25
>
> **Nits**
> > "Coarse-to-fine" was perhaps not the best terminology.
>
> We thank the reviewer for pointing out this issue. We also recognize that the current Coarse-to-Fine terminology may not be appropriate, and we will revise and correct it in the updated manuscript.
>
> > Figure 1 - Why no arrow from (4) to (5)?
>
> Because the task execution and trajectory evaluation is decoupled. The collected trajectories during task execution are first saved in center machich, and then automated evaluation is conducted on collected trajectories.

---

### Note · Authors · 2026-01-04

I have read and agree with the venue's withdrawal policy on behalf of myself and my co-authors.